# Lhx1 maintains synchrony among circadian oscillator neurons of the SCN

Megumi Hatori[1][\*][†][‡], Shubhroz Gill[1][†], Ludovic S Mure[1], Martyn Goulding[2], Dennis D M O'Leary[2], Satchidananda Panda[1][\*]

[1]Regulatory Biology Laboratory, Salk Institute for Biological Studies, La Jolla, United States; [2]Molecular Neurobiology Laboratory, Salk Institute for Biological Studies, La Jolla, United States

**Abstract** The robustness and limited plasticity of the master circadian clock in the suprachiasmatic nucleus (SCN) is attributed to strong intercellular communication among its constituent neurons. However, factors that specify this characteristic feature of the SCN are unknown. Here, we identified *Lhx1* as a regulator of SCN coupling. A phase-shifting light pulse causes acute reduction in *Lhx1* expression and of its target genes that participate in SCN coupling. Mice lacking *Lhx1* in the SCN have intact circadian oscillators, but reduced levels of coupling factors. Consequently, the mice rapidly phase shift under a jet lag paradigm and their behavior rhythms gradually deteriorate under constant condition. Ex vivo recordings of the SCN from these mice showed rapid desynchronization of unit oscillators. Therefore, by regulating expression of genes mediating intercellular communication, Lhx1 imparts synchrony among SCN neurons and ensures consolidated rhythms of activity and rest that is resistant to photic noise.

**\*For correspondence:** mhatori@
a6.keio.jp (MH); panda@salk.edu
(SP)

[†]These authors contributed
equally to this work

**Present address:** [‡]School of
Medicine, Keio University, Tokyo,
Japan

**Competing interests:** The
authors declare that no
competing interests exist.

**Reviewing editor**: Louis Ptáček,
University of California, San
Francisco, United States

## Introduction

Circadian clocks generate ~24 hr rhythms in behavior and physiology which allow an organism to anticipate and adjust to environmental changes accompanying the earth's day/night cycle. These rhythms are generated in a cell-autonomous manner by transcription–translation based feedback loops which are composed of clock proteins, such as PERIOD (PER1, PER2, and PER3), CRYPTOCHROME (CRY1 and CRY2), CLOCK, BMAL1, REV-ERB (REV-ERBα and REV-ERBβ), and ROR (RORα, RORβ, and RORγ) in mammals (*Mohawk et al., 2012*). These oscillatory loops reside in almost all tissue types and regulate their downstream effectors to generate oscillations in the steady-state mRNA levels of thousands of genes in a tissue-specific manner. Identification of tissue-specific circadian transcripts in peripheral organs has elucidated the mechanism by which circadian clocks dictate the temporal regulation of organ function. The tissue level clocks are organized in a hierarchical manner. The hypothalamic suprachiasmatic nucleus (SCN) composed of ~20,000 densely packed neurons acts as the master clock by orchestrating molecular oscillations in peripheral tissues (*Welsh et al., 2010*). Tight intercellular communication among SCN neurons (coupling) drives synchronous oscillations. This, in turn, imparts overt rhythms in activity-rest and dependent rhythms in physiology and metabolism of the whole organism. While coupling between the SCN neurons buffers against the noise in oscillations of the constituent neurons, it is plastic enough to allow adaptive resetting of the phase of the SCN oscillator in response to changes in the environment.

Light is the principal cue for entraining the SCN circadian clock to environmental cycles. Light stimuli are perceived in the retina and transmitted to the SCN via melanopsin (OPN4)-expressing retinal ganglion cells (mRGCs) (*Hatori and Panda, 2010*). The time-of-the-day specific response (called 'gating') of the SCN to light pulses properly adjusts the phase of the circadian clock. In mice held under constant darkness, light administered at subjective daytime, subjective evening, or subjective

**eLife digest** As anyone who has experienced jet lag can testify, our sleeping pattern is normally synchronized with the local day–night cycle. Nevertheless, if a person is made to live in constant darkness as part of an experiment, they still continue to experience daily changes in their alertness levels. In most individuals, this internal 'circadian rhythm' repeats with a period of just over 24 hr, and exposure to light brings it into line with the 24-hr clock.

The internal circadian rhythm is generated by a structure deep within the brain called the suprachiasmatic nucleus (SCN), which is essentially the 'master clock' of the brain. However, each cell within the SCN also contains its own clock, and can generate rhythmic activity independently of its neighbors. Cross-talk between these cells results in the production of a single circadian rhythm.

Now, Hatori et al. have identified the master regulator that controls this cross-talk. When mice living in 24-hr darkness were exposed to an hour of light in the early evening, they showed changes in the levels of proteins associated with many SCN genes. But one gene in particular, known as *Lhx1,* stood out because it was strongly suppressed by light.

Mice with a complete absence of *Lhx1* die in the womb. However, mice that lose *Lhx1* during embryonic development survive, although they struggle to maintain circadian rhythms when kept in complete darkness. This is not because their SCN cells fail to generate circadian rhythms. Instead, it is because the loss of *Lhx1*—a transcription factor that controls the expression of many other genes—means that the SCN cells do not produce the proteins they need to synchronize their outputs.

As well as identifying a key gene involved in the generation and maintenance of circadian rhythms, Hatori et al. have underlined the importance of cell-to-cell communication in these processes. These insights may ultimately have therapeutic relevance for individuals with sleep disturbances caused by jet lag, shift work or certain sleep disorders.

late night causes no shift, phase delay or phase advance of the behavioral rhythm respectively. This phase-dependent light response is conserved across species (*Pittendrigh, 1967*; *Zatz et al., 1988*; *Schwartz and Zimmerman, 1990*). It is known that illumination at night triggers extensive chromatin remodeling in the mouse SCN (*Crosio et al., 2000*), which likely results in changes in the levels of a large number of transcripts. However, except for a few dozen transcripts including the clock genes *Per1* and *Per2* (*Zhu et al., 2012*; *Jagannath et al., 2013*), the extent of light-triggered transcriptional changes in the SCN is largely unknown. These acute transcriptional changes impinge on the molecular oscillator to adjust the phase of the mRNA rhythms in the SCN.

It is becoming increasingly clear that coupling among the SCN neurons buffers against phase shifts, and the transient weakening of such coupling facilitates large phase shifts. SCN neurons exhibit tight intercellular communication imposed by paracrine peptidergic signals such as VIP (Vasoactive intestinal polypeptide), AVP (Arginine vasopressin), and GRP (Gastrin-releasing peptide) (*Welsh et al., 2010*; *Hogenesch and Herzog, 2011*). As the deficiency of *Vip* or its receptor *Vpac2r* causes desynchronization among SCN neurons (*Aton et al., 2005*), this peptide-mediated coupling mechanism is a unique and necessary feature of the SCN in order to generate robust synchronous rhythms. Weaker coupling among the SCN neurons is suggested to facilitate rapid and large phase shifts of the overt rhythms (*Herzog 2007*; *An et al., 2013*). Therefore, transcription factors that regulate expression of the SCN coupling agents are central to the unique function of the SCN.

Here, using a combination of behavioral, genetic, and genomic tools, we vastly expand the understanding of the dynamic transcriptional landscape of the SCN. We conducted comprehensive analysis of the light-regulated, circadian, and tissue-enriched protein-coding transcriptome of the mouse SCN in order to understand the specificity of the master circadian clock. We found that the SCN-enriched transcription factor Lhx1 (LIM homeobox 1) is required for expression of a number of genes including *Vip* whose protein product participates in intercellular signaling. The SCN-specific loss of *Lhx1* attenuates cell to cell coupling of cell-autonomous oscillators in the SCN, abolishing circadian behavioral activity consolidation in vivo.

# Results

## Light-regulated and circadian-controlled transcripts of the SCN

To comprehensively identify circadian-, light-regulated, and SCN-enriched protein-coding transcripts in the adult SCN, following 2 weeks of entrainment to 12 hr light:12 hr dark (LD) cycles male C57BL6/J mice were transferred to constant darkness (DD), and the SCN was collected every 2 hr over 48 hr. Light at subjective night, but not during subjective day is known to cause a behavioral phase shift. To assess the gene expression effect of a phase-shifting pulse of light, a subset of mice were exposed to a 1 hr white light pulse delivered at 30 hr, 40 hr, and 46 hr after the onset of DD (*Figure 1A*, *Figure 1—figure supplement 1*) representing subjective daytime (CT6), early evening (CT16), and late night

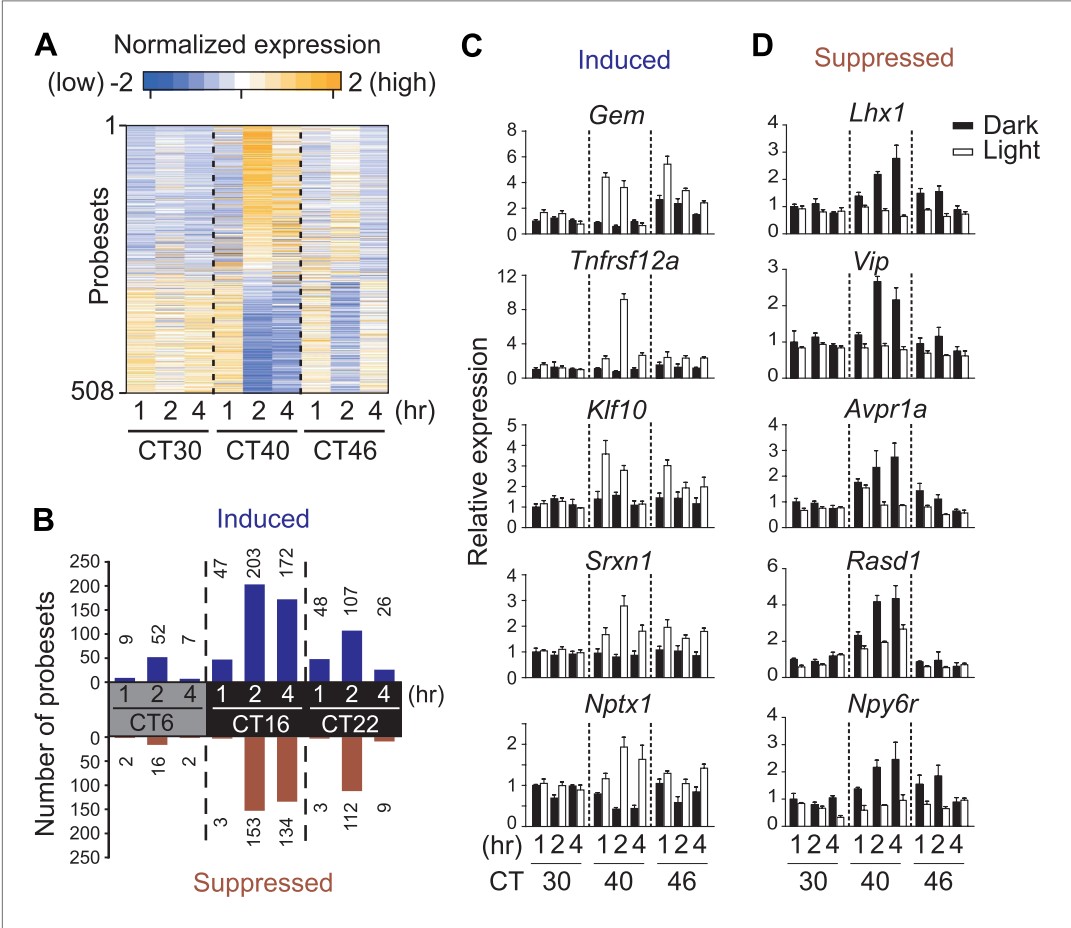

**Figure 1**. Light-regulated transcripts of the SCN. (**A**) Heatmap rendering of light-regulated SCN transcripts. For each time point, fold change between respective light treated and dark control was plotted. (**B**) Circadian gating of light-modulated transcripts. Cutoffs of two fold were set for up-regulation (blue) or suppression (red) after light pulse, and the number of probesets that satisfy each cutoff was plotted for each point. Quantitative RT-PCR (qRT-PCR) expression confirmation of genes detected as light-regulated by microarray. Examples of genes (**C**) induced or (**D**) repressed by light pulses at three different time points. (Mean +s.e.m., n = 4).

The following figure supplements are available for figure 1:

**Figure supplement 1**. Transcriptional profiling of the mouse SCN.

**Figure supplement 2**. Light-induced changes in SCN gene expression correlate with the known effect of light on phase shift in different genetic models of light signaling.

**Figure supplement 3**. SCN enriched (not SCN-exclusive) transcripts.

(CT22). We discovered 1412 genes (*Figure 1—figure supplement 1*; *Supplementary file 1*) that show circadian oscillation in transcript abundance (pMMCβ <0.05, pFGT <0.05, and median temporal expression >100). Gene expression measured at 1 hr, 2 hr, and 4 hr after the beginning of the 1 hr light pulse revealed 508 probesets whose levels changed (up- or down-regulated) in response to at least one of the three light pulses (*Figure 1A,B*, *Figure 1—figure supplement 1*; *Supplementary file 2A*). However, only 84 (17%) were also rhythmic (*Figure 1—figure supplement 1*; *Supplementary file 2B*).

We tested whether the number of light-modulated transcripts parallels the phase shifting effect of light. Light pulse delivered during subjective day (CT6) triggered changes in a small number of transcripts, while the same pulse at CT16 causes a large phase shift accompanied by a large number of transcripts changing $\geq 2$ fold (*Figure 1B*, *Figure 1—figure supplement 1*; *Supplementary file 2C*). The magnitude of light modulation of transcripts also paralleled the known phase shifting effect of light in genetic models of light input perturbation. Compared to wild type (WT) C57BL6/J, the light response was unchanged in mice lacking rod/cone photoreceptors, attenuated in melanopsin (*Opn4*)-deficient mice and completely abolished in mice lacking all three photopigments or those with specific ablation of melanopsin expressing retinal ganglion cells (*Opn4$^{Cre/+}$;R26$^{iDTR/+}$* + diphtheria toxin [DT]) (*Figure 1—figure supplement 2*). Overall, the light-induced transcriptional responses in the SCN closely correlate with the extent of light-induced behavioral phase shift (*Hatori and Panda, 2010*) and hence are informative of the mechanism and consequences of the phase shift.

The light-induced genes include the circadian clock components e.g., *Per1*, *Per2*, and *Bhlhe40* (*Dec1*) and genes involved in the CREB and MAPK signaling pathways (*Zhu et al., 2012*; *Jagannath et al., 2013*), which are implicated in intracellular signaling leading to resetting the phase of cell autonomous oscillators (*Figure 1C*). The novel group of light repressed transcripts was enriched for those involved in inter-cellular communication including *Vip*, *Avpr1a*, *Rasd1*, and *Npy6r* (*Figure 1D*, *Supplementary file 2C*), suggesting that an effective phase shift of the SCN clock rests both on resetting the phase of cell autonomous clock and on light-induced relaxation of the intercellular coupling.

The data for circadian and light-dependent gene expression are also being made available in a user friendly searchable web interface at **http://scn.salk.edu**. The database can be queried using a gene symbol or a probeset identifier as the keyword.

## Lhx1 is an SCN-enriched and light-modulated gene

Since the tight intercellular coupling of circadian oscillators is largely an SCN specific phenomenon (*Herzog 2007*), we reasoned that light likely suppresses the expression of an SCN-enriched factor that coordinates intercellular communication. We employed three-step enrichment criteria comparing the transcriptome of the SCN with that of 82 other mouse tissues including 14 different neural tissues (*Su et al., 2004*) (*Figure 1—figure supplement 3*). This analysis identified 213 SCN-enriched (not SCN-exclusive) genes (*Figure 1—figure supplement 3*, *Supplementary file 3*), including *Rorα*, *Rorβ*, *Vip*, *Grp*, *Rgs16,* and *Prokr2* which are known to play important roles in SCN function (*Welsh et al., 2010*; *Doi et al., 2011*; *Kasukawa et al., 2011*; *VanDunk et al., 2011*).

Among the 13 SCN-enriched transcription factors discovered (*Figure 1—figure supplement 3*), *Lhx1* mRNA was suppressed by light (*Figure 1D*), raising the possibility that it regulates the expression of SCN synchronizing agents and that the lack of *Lhx1* might render the SCN prone to desynchrony. However, *Lhx1* is a necessary factor for differentiation of several tissue types as *Lhx1$^{-/-}$* embryos die ~ E10 (*Shawlot and Behringer, 1995*). In the hypothalamus, *Lhx1* expression begins at E11 and parallels that of *Six6* (*VanDunk et al., 2011*), whose function is necessary for normal SCN development (*Clark et al., 2013*). Since *Rorα* expression is also SCN enriched (*Figure 1—figure supplement 3*) and its developmental expression follows *Lhx1* expression in the SCN region, we generated *Rorα$^{Cre}$;Lhx1$^{loxP}$* mice for testing the role of *Lhx1* in SCN function.

## Loss of Lhx1 in the SCN affects circadian consolidation and light induced phase shifts of behavioral rhythms

The overlap between *Rorα* and *Lhx1* expression in the adult brain is largely restricted to the SCN (*Figure 1—figure supplement 3*) and the dLGN (*Chou et al., 2013*). The Rorα/Lhx1 double positive dLGN neurons constitute the thalamocortical pathway for conveying visual information to the brain (*Chou et al., 2013*). Since this circuit is not implicated in circadian photoentrainment (*Mure and Panda, 2012*), we reasoned the *Rorα$^{Cre}$;Lhx1$^{loxP}$* mice will be appropriate for testing the role of Lhx1 in SCN.

Rorα is an essential component of the cell autonomous circadian oscillator (*Sato et al., 2004*). Its mRNA becomes detectable from E15 onward (*VanDunk et al., 2011*). Therefore, in the early developmental stage between E11 and E15, *Lhx1* is expected to be expressed in the *Rorα^Cre;Lhx1^loxP* double mutant mice, which would allow normal Lhx1 function (if any) during early SCN differentiation, while uncovering the post-developmental role of Lhx1 in SCN function.

The *Rorα^Cre* mouse has an IRES;Cre cassette knocked-in downstream of the *Rorα* locus (*Chou et al., 2013*) which permits normal expression of *Rorα* and co-expression of Cre. *Rorα^Cre;R26R* mice (*Figure 2A* and *Figure 2—figure supplement 1*) or *Rorα^Cre;Z/AP* mice (*Figure 2B* and *Figure 2—figure supplement 1*) showed robust *Cre*-dependent LacZ or ALPP (alkaline phosphatase) expression in the SCN region of the hypothalamus. ALPP staining of *Rorα^Cre;Z/AP* mice revealed uniform staining of SCN neurons along dorso-ventral and rostro-caudal axes (*Figure 2—figure supplement 1*). As opposed to the developmental and circadian dysfunction in *Rorα* mutant (*Sato et al., 2004*), the *Rorα^Cre* allele did not compromise Rorα function as the *Rorα^Cre* mice exhibited normal development and circadian activity rhythm in light–dark cycles and in constant darkness that are indistinguishable from those of wild-type mice (see below).

A single copy of *Cre* in *Rorα^Cre/+:Lhx1^loxP/loxP* mice reduced *Lhx1* mRNA levels by nearly 40%, while in *Rorα^Cre/Cre:Lhx1^loxP/loxP* (*Lhx1^SCN-KO*) mice *Lhx1* expression in the SCN is severely reduced (*Figure 2D*). The gross morphology of the SCN in these mice remains intact (*Figure 2—figure supplement 1*) suggesting that the conditional loss of *Lhx1* post early development in the SCN does not severely affect its differentiation unlike the loss of *Six3, Six6*, or *Math5* (*Wee et al., 2002*; *VanDunk et al., 2011*; *Clark et al., 2013*). The activity pattern of these mice entrains normally to an imposed LD cycle (*Figure 2—figure supplement 2*) implying functional innervation of the SCN by the mRGCs, which is known to occur postnatally (*McNeill et al., 2011*). Accordingly, anterograde labeling using a Cholera Toxin B conjugated fluorescent marker indicated normal innervation of the SCN by RGCs (*Figure 2E,F* and *Figure 2—figure supplement 1*).

Light-induced upregulation of immediate early genes and clock components including *c-Fos, JunB, Per1*, and *Per2* in the SCN of *Lhx1^SCN-KO* mice was comparable to that in WT mice (*Figure 2G*). However, light suppressed transcripts involved in intercellular communication such as *Vip* and *Avpr1a* showed reduced basal expression under DD (*Figure 2H*). This acute induction of the phase-resetting branch of light input along with potentially weak intercellular coupling suggested that the *Lhx1^SCN-KO* mice may be more susceptible to light-induced phase shifts. Accordingly, in response to an 8 hr phase advance or delay of the LD cycle, the activity onset of *Rorα^Cre/+;Lhx1^loxP/loxP* and *Rorα^Cre/Cre;Lhx1^loxP/loxP* mice readjusted much faster than the WT mice to the new LD regime irrespective of the direction of the shift (*Figure 2I–L*).

Next, we tested the consequence of the potentially weak intercellular communication in the absence of light. Under constant darkness (DD), the circadian locomotor activity rhythms of the *Rorα^+/+; Lhx1^+/+*, *Rorα^+/+;Lhx1^loxP/loxP, Rorα^Cre/+;Lhx1^+/+*, or *Rorα^Cre/Cre;Lhx1^+/+* mice were comparable (*Figure 3A–D, Table 1*). The *Rorα^Cre/+;Lhx1^loxP/loxP* mice showed normal activity rhythm for up to 3 weeks under DD, after which the activity consolidation deteriorated with no apparent ~24 hr rhythm (*Figure 3E*). The *Rorα^Cre/Cre;Lhx1^loxP/loxP* mice showed circadian activity rhythm for up to 4 days under constant darkness, after which they became arrhythmic (*Figure 3F, Figure 3—figure supplement 1*). The lack of circadian locomotor activity rhythm in *Lhx1^SCN-KO* mice under DD does not result from the disruption of the cell autonomous circadian oscillator, since the median expression of core clock components *Per1* and clock output gene *Dbp* largely remained equivalent in the SCN of *Lhx1^SCN-KO* and wild-type cohorts (*Figure 3G,H*). Both transcripts showed a significantly dampened rhythm with reduced peak levels and increased expression at the trough, suggestive of oscillator desynchrony. On the other hand, transcripts participating in intercellular communication including *Vip, Avpr1a, Rasd1, Pde7b, Creb3l1*, and a cell matrix associated cell–cell interaction mediator *Nov* were significantly reduced in the *Lhx1^SCN-KO* mice (*Figure 3I*).

## Lhx1 regulates expression of Vip

Among the downregulated genes, the mRNA levels of *Vip* were undetectable in the Lhx1^SCN−KO mice (*Figure 3J*), and *VIP* protein level was also reduced in the SCN (*Figure 3K*). Hence, we tested whether LHX1 regulates *Vip* expression. Transcription from a 1 kb promoter region of mouse *Vip* driving a luciferase reporter was activated by wild-type mouse and human Lhx1 in a dose-dependent manner (*Figure 3L* and *Figure 3—figure supplement 2*). Such transcriptional regulation was dependent on the DNA binding function of LHX1 because the LHX1^N230S mutant failed to activate transcription from *Vip:luc*

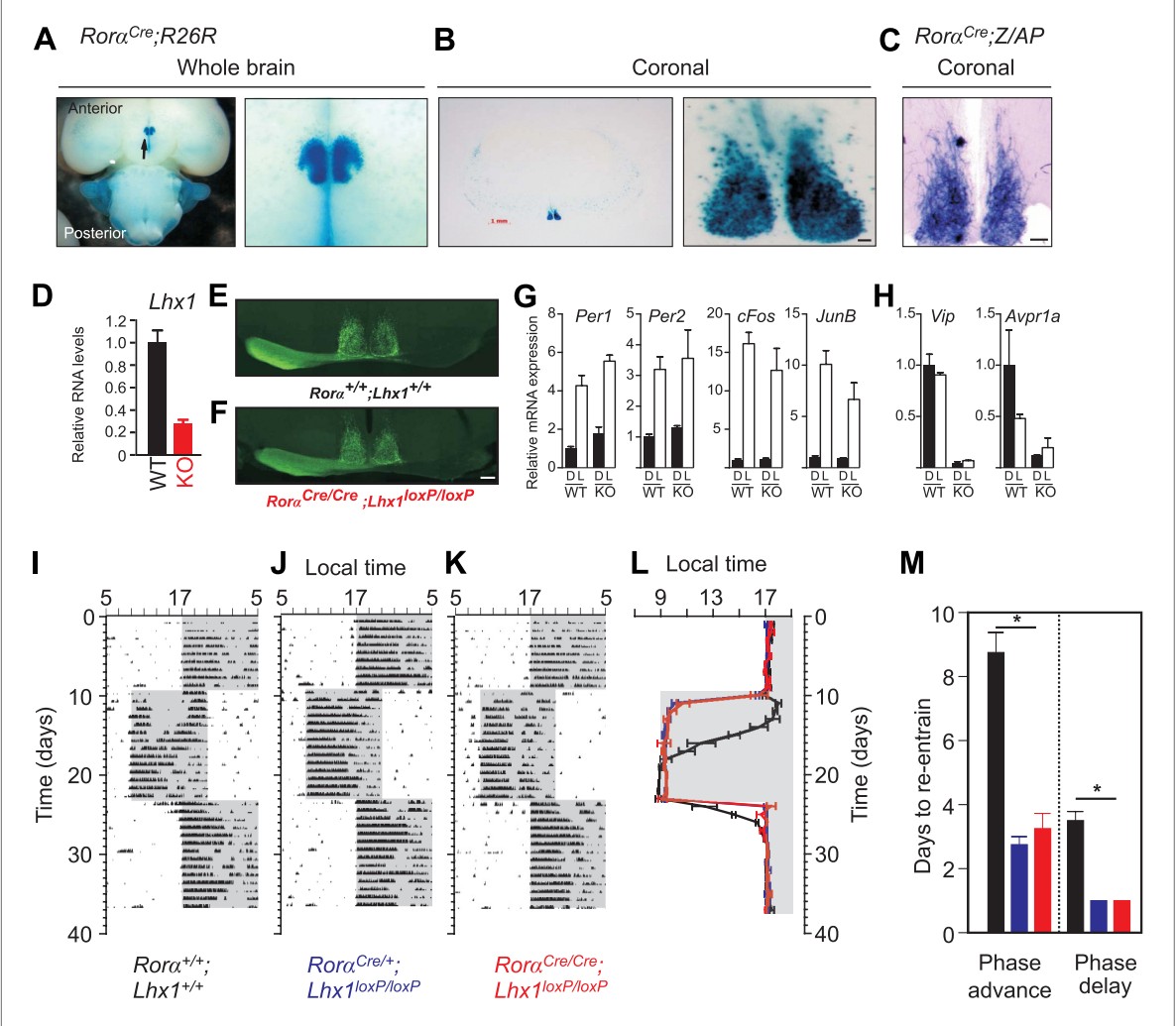

**Figure 2**. Loss of Lhx1 expression in the SCN renders faster synchronization with change in LD regimes. Enriched expression of a *Rorα*-driven marker in the SCN in *Rorα^Cre;R26R* mice. (**A**) Ventral view of a whole brain (magnified view on the right) of adult *Rorα^Cre;R26R* shows LacZ staining of the SCN. (**B**) Coronal section through the mid-SCN region (scale bar, 1 mm) and the magnified view of the SCN (scale bar, 100 μm) showing LacZ expression or (**C**) alkaline phosphatase expression in *Rorα^Cre;Z/AP* mice. (**D**) qRT-PCR estimate of Lhx1 expression in the SCN (mean +s.e.m, n = 5). (**E**) Normal SCN innervation of the retinal ganglion cells in the WT mice as revealed by monocular injection of CTB-conjugated fluorescent marker is intact in (**F**) Lhx1^SCN–KO mice. A 1 hr light pulse at CT16 causes (**G**) upregulation of light-induced genes (*Per1*, *Per2*, *cFos*, JunB), while (**H**) the light-suppressed transcripts (*Lhx1*, *Vip*, *Avpr1a*) in the WT SCN show reduced expression in the *Lhx1^SCN-KO* mice. Mice were in DD for 2 days before the light pulse. Representative actograms of (**I**) *Rorα^Cre/Cre*, (**J**) *Rorα^Cre/+;Lhx1^loxP/loxP*, and (**K**) *Rorα^Cre/Cre;Lhx1^loxP/loxP* mice subjected to 8 hr phase advance and 8 hr delay. (**I**) Average (+s.e.m., n = 5–8) activity onset and (**K**) average (+s.e.m.) number of days to re-entrain to advance or delay in light onset in three genotypes. Color codes in **L** and **M** correspond to the labels in **I**–**K**.

The following figure supplements are available for figure 2:

**Figure supplement 1**. Histology of the adult SCN.

**Figure supplement 2**. Activity profile under light-dark condition.

reporter (**Figure 3L**). This mutant (**Figure 3—figure supplement 2**) carries a missense mutation in a highly conserved asparagine residue known to disrupt DNA binding (**Thaler et al., 2002**). Furthermore, wild-type LHX1 also failed to activate transcription from *SV40* promoter, supporting the idea that *Vip* expression is specifically activated by LHX1 and that this requires the DNA-binding activity of LHX1. Collectively, these results demonstrate that LHX1 regulates expression of *Vip*, which along with sufficient expression of other synchronizing agents maintains synchrony among SCN neurons.

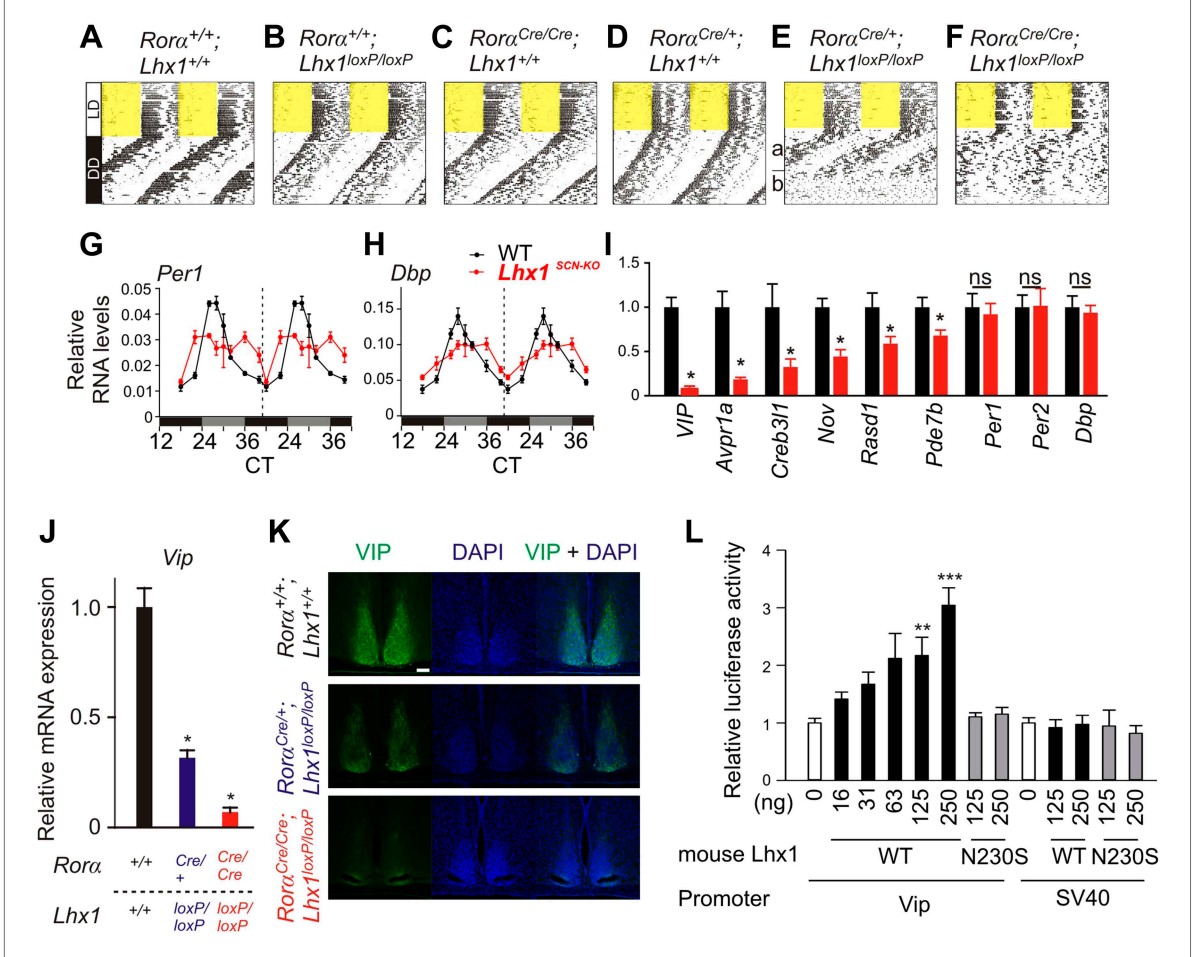

**Figure 3**. Lhx1 sustains normal circadian activity rhythms by regulating expression of synchronizing factors. (**A–F**) Representative wheel running activity pattern over several days of LD followed by DD in wild type and mice lacking *Lhx1* in the SCN. Double-plotted qRT-PCR quantification (average +s.e.m, n = 3–4 mice) of (**G**) *Per1* and (**H**) *Dbp* in the SCN of DD adapted WT and Lhx1[SCN–KO] mice. (**I**) Average (+s.e.m. 8 time points every 3 hr over 24 hr) expression of several factors involved in intercellular communication or circadian clock in the SCN of dark adapted WT and Lhx1[SCN–KO] mice. (**J**) Average mRNA (+s.e.m., n = 3–6 mice, *p < 0.05) expression or (**K**) immunoreactivity of VIP is reduced in the SCN of Lhx1-deficient mice. (**L**) Transcriptional activation of mouse *Vip* promoter by mouse LHX1. pGL3-promoter vector was used as a control promoter vector. Values are mean +s.e.m, ANOVA **p<0.01, ***p<0.001 vs 0 ng (white bar).

The following figure supplements are available for figure 3:

**Figure supplement 1**. Activity profile under constant darkness.

**Figure supplement 2**. Lhx1 activates Vip transcription.

## Lhx1 mediates synchrony among SCN oscillator neurons

Coupling among SCN neurons supports synchronous oscillations of individual oscillators that otherwise show variation in intrinsic period lengths (*Aton et al., 2005*). To directly test the role of Lhx1 in maintaining SCN synchrony, we recorded electrophysiological activity of SCN slices using a multielectrode array. SCNs of both WT and *Lhx1[SCN-KO]* mice showed no apparent difference in the firing frequency, supporting the notion that *Lhx1[SCN-KO]* has no discernible developmental defects in the SCN. The WT SCN from LD reared animals showed synchronous firing rhythm, as characterized by both robust oscillation of multiunit activity as well as synchrony of peak phase of activity among different channels. The *Lhx1[SCN-KO]* SCN, on the other hand, showed remarkable dampening of firing rhythm as well as phase dispersion on day 1 and became almost asynchronous within 3 days (*Figure 4A–E*). This

**Table 1.** Circadian running wheel activity period length of various mouse strains under constant darkness (n = 7–22)

| | Average (h) | SEM (h) |
|---|---|---|
| *Rora+/+;Lhx1+/+* | 23.78 | 0.23 |
| *Rora+/+;Lhx1loxP/loxP* | 23.64 | 0.05 |
| *RoraCre/Cre;Lhx1+/+* | 23.40 | 0.14 |
| *RoraCre/+;Lhx1+/+* | 23.86 | 0.07 |
| *RoraCre/+;Lhx1loxP/loxP* | 23.51 | 0.13 |
| *RoraCre/Cre;Lhx1loxP/loxP* | NA | NA |

Mice showing arrhythmic activity were excluded from the analysis. (NA = Not Applicable).

parallels the timeline of emergence of behavioral arrhythmicity in *Lhx1SCN-KO* mice when they are released from LD to DD (*Figure 3F*). Similarly, in WT mice maintained in DD for 2 weeks, the SCN displayed circadian oscillation of the multi-unit activity with a similar peak phase among the different electrodes examined (*Figure 4—figure supplement 1*). In contrast, the slices from *RoraCre/Cre;Lhx1loxP/loxP* animals showed widely dispersed phases of the multi-unit activity peak, a finding coherent with the arrhythmic locomotor activity observed in these animals in DD (*Figure 3A*). Since VIP expression is severely reduced in the *Lhx1SCN-KO* mice, we tested if extrinsic supplementation of VIP can restore the synchrony of the SCN neurons. Daily application of VIP for 1 hr on SCN slices from DD adapted *Lhx1SCN-KO* mice restored the normal synchrony and waveforms of the SCN firing rhythm (*Figure 4F,G*). Finally, the gradual dampening of SCN multiunit activity and desynchrony in the LD-adapted SCN slice from *Lhx1SCN-KO* mouse can be reversed by the daily application of VIP (*Figure 4H,I*).

## Discussion

Tissue-specific gene expression and temporal changes in transcript levels govern and facilitate tissue function. Although the SCN has long been recognized as the master circadian oscillator and the principal target for light modulation of circadian rhythms in mammals, the molecular basis for these at the protein-coding transcript level have not been comprehensively identified. In this study, we present a thorough analysis of transcriptional oscillations in the SCN, its response to light at different phases of the circadian oscillator and a glossary of SCN-enriched genes. This detailed description of the transcriptional landscape of the SCN has a variety of implications for understanding SCN function. In this study, we focused on the relevance of oscillator synchrony in circadian rhythms and light-induced phase shift in behavior.

Under natural conditions, time-of-the-day dependent interaction of the SCN with light input from the retina specifies the phase of overt activity-rest rhythms (*Hatori and Panda, 2010*). Light during the subjective night induces expression of several genes including immediate early genes *c-fos*, activates MAPK pathway, acutely upregulates the *Per1* transcript and resets the phase of overt behavioral rhythm (*Obrietan et al., 1998*, *1999*; *Dziema et al., 2003*; *Cheng et al., 2004*; *Butcher et al., 2005*). We found that global transcriptional changes in the SCN parallel the sensitivity of the activity rhythms to light. During the subjective day, when light is ineffective in resetting overt rhythms, a very small fraction of the transcriptome changed in response to light. Conversely, during the subjective night, extensive changes in transcripts correlated with the large phase shifting effect of light, suggesting that changes in multiple pathways accompany the shift in the phase of the SCN oscillator network. Among the transcripts that change both during daytime and nighttime, their magnitude of expression was attenuated during the daytime. The magnitude of transcript changes also reflected the effectiveness of retinal photoreceptors in entraining the clock. As seen in the *Opn4−/−* mice (*Panda et al., 2002b*), attenuated light-induced change in overt rhythms paralleled the reduced transcriptional responses to light. Mice lacking mRGCs or those lacking rod, cone and melanopsin photopigments exhibited no behavioral phase shift in response to light (*Hatori and Panda, 2010*). They also showed no significant mRNA expression changes in transcripts in the SCN (*Figure 1—figure supplement 2*). Collectively, these observations support the notion that a light signal perceived through retina photoreceptors causes transcriptional changes in the SCN that parallel the behavioral phase-shifting effect of light.

Included among the light-induced transcripts were several components of intracellular signaling cascades, kinases, phosphatases, and transcription factors (MAPK pathway genes, SIK1, PER1, PER2, DEC1, EGR1, and EGR2), which likely function at the different signaling steps needed to transduce light information received at the plasma membrane to generate an appropriate change in the phase of the core clock (*Obrietan et al., 1998*, *1999*; *Dziema et al., 2003*; *Cheng et al., 2004*;

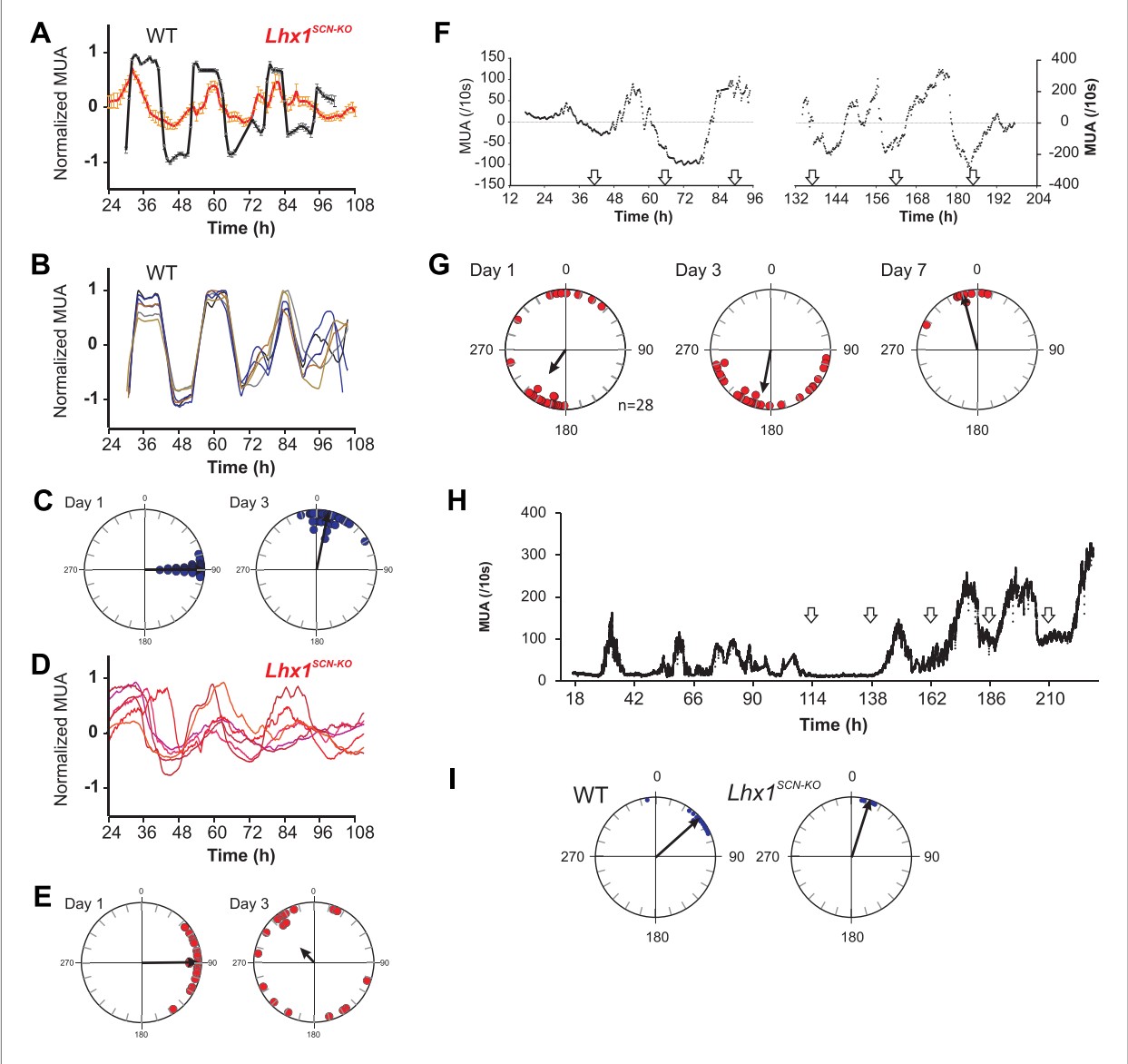

**Figure 4**. Lhx1 maintains synchrony among SCN neurons partly via VIP. (**A**) Average (+s.e.m.) normalized multiunit activity (MUA) recorded from representative SCN slices of LD-adapted WT (n = 40, black) and Lhx1SCN–KO (n = 12, orange) mice. Data were binned every 60 min. Representative normalized MUAs and peak phase of activity from WT SCN (**B** and **C**, n = 40) and Lhx1SCN–KO mouse (**D** and **E**, n = 19). For **C** and **E**, left and right panels are respectively for days 1 and 3. (**F**) Average MUA of a DD adapted Lhx1SCN–KO SCN that received 1 hr perfusion of VIP daily for up to 7 days. (**G**) Peak phases of activity are gradually synchronized over 7 days. (**H**) Representative MUA from the SCN of an LD-adapted Lhx1SCN–KO mouse over several days. During the first 4 days, the activity dampened, which was rescued by daily application of VIP. Down arrows in **F** and **H** indicate the time of VIP application. (**I**) Peak phases of activity in WT and Lhx1SCN–KO SCN at the end of 7 days are shown.

The following figure supplement is available for figure 4:

**Figure supplement 1**. Normalized multi-unit activity recorded from DD adapted WT and Lhx1SCN-KO (mean ± SEM).

*Butcher et al., 2005*; *Jagannath et al., 2013*). The light-suppressed transcripts on the other hand included intercellular signaling agents, cell surface receptors, and associated signaling components. This indicated that stimulus-induced downregulation of the tight coupling among SCN neurons is a likely mechanism to allow desynchronization and then resynchronization of cellular clocks to reset the SCN to a new phase. Since tight intercellular communication among the SCN neurons is a characteristic feature of the SCN, we further focused on the subset of transcripts that are light-suppressed as well

as enriched in the SCN. The search for SCN-specific genes among 82 different mouse tissues did not yield any protein coding gene that is exclusively expressed in the SCN, suggesting that a combinatorial gene expression signature specifies SCN development and function. LHX1 is one of the SCN-enriched and light-suppressed transcription factors, and hence we reasoned that LHX1 might regulate transcription of at least a subset of the intercellular communication agents that specify SCN neuronal network features.

During the preparation of this manuscript, another study showed *Six3-Cre* dependent loss of *Lhx1* expression severely affected terminal differentiation as well as peptidergic outputs of the SCN (*Bedont et al., 2014*). By using *Rora-Cre* in our study, the loss of *Lhx1* expression likely occurred after the onset of *Rorα* expression at E14.5, thereby allowing any terminal differentiation function of *Lhx1* in SCN development to occur normally. In both studies, the loss of Lhx1 in the SCN did not affect the overall expression level of any of the clock components, while the expression of several genes implicated in cellular synchrony including *Vip* and *Avpr1a* were downregulated. AVP, VIP, and their cognate receptors have been implicated in maintaining inter-cellular communication among SCN neurons (*Herzog, 2007*). Light pulse also reduced the expression of *Avpr1a* and *Vip* mRNA. Mice carrying loss-of-function alleles of *Vip* or vasopressin receptors exhibit a weakened SCN network, progressive desynchronization of SCN neurons and increased sensitivity to light-induced phase shifts (*Herzog, 2007*; *Yamaguchi et al., 2013*). Similarly, the loss of AVP receptor weakens the network and increases the sensitivity of the behavioral rhythm to changes in the light regime (*Herzog, 2007*). While these observations have indicated that AVP- and VIP-mediated intercellular communication constitutes the framework for the SCN network, the transcription factor(s) that determines this SCN-specific property was unknown. Direct transcriptional induction from the VIP promoter by LHX1, the severe loss of VIP mRNA and immunoreactivity in the *Lhx1*$^{-/-}$ SCN along with a desynchronous SCN now establishes Lhx1 as a critical regulator of VIP production in the SCN.

In summary, we have discovered that Lhx1 is a master regulator of multiple factors including VIP that maintain robust coupling among SCN neurons after their differentiation into oscillator neurons. While the loss of *Vip* (*Colwell et al., 2003*), *Avpr1a* (*Wersinger et al., 2007*; *Li et al., 2009*), or *Rasd1* alone causes a mild alteration in the circadian organization or light responses (*Cheng et al., 2004*), the parallel perturbation of multiple intercellular signaling components in the *Lhx1*$^{SCN-KO}$ mice indicates a critical role for Lhx1 in determining the specific feature of SCN neurons that impart coupling among neurons. The intercellular coupling is thus as important as the cell-autonomous oscillations for maintaining the consolidated rhythm of activity-rest that can resist abrupt changes in the ambient light conditions.

## Materials and methods

### Animals

All animal experiments were carried out in accordance with the guidelines of the Institutional Animal Care and Use Committee of the Salk Institute. Mice were housed under 12 hr light: 12 hr dark (LD) cycles. Food and water were available *ad libitum*. C57BL/6J, C3H/HeJ strain (*rd*) carrying *Pdeb*$^{rd1}$ mutation, Cre-dependent lacZ reporter strain (*R26R*) (*Soriano, 1999*), and Cre-dependent human ALPP reporter strain (*Z/AP*) (*Lobe et al., 1999*) were obtained from the Jackson Laboratory. *Opn4*$^{-/-}$ mice (*Panda et al., 2002b*) were bred to *rd/rd* to generate *rdrd;Opn4*$^{-/-}$. Both *Opn4*$^{Cre}$ and *Opn4*$^{Cre}$;*R26*$^{iDTR}$ were described in *Hatori et al. (2008)*. The floxed *Lhx1* allele (*Lhx1*$^{loxP}$) mice were originally generated in *Kwan and Behringer (2002)*. *Rorα*$^{Cre}$ mouse was generated by knocking in an IRES;Cre cassette 3′ downstream of the *Rorα* locus (*Chou et al., 2013*). Both *Lhx1*$^{loxP}$ and *Rorα*$^{Cre}$ mice were back-crossed to C57BL/6J strain for at least eight generations.

### DNA microarrays

168 male C57BL/6J mice of 6 weeks of age were maintained for 3 weeks on a 12 hr light:12 hr dark cycle. For circadian gene expression profiling, after being placed in DD for 2 days, four animals were sacrificed every 2 hr, beginning at hour 30 of DD, which corresponds to CT18, for two complete 24 hr cycles (*Figure 1A*). For light-regulated gene profiling, mice were maintained in DD then exposed to 1 hr light at CT30, 40, or 46, while control (no light pulse) mice were left in dark. After 1 hr, all mice were returned to DD, and four animals each were collected at 1, 2, and 4 hr from the beginning of light exposure from each CT. Mice were sacrificed by cervical dislocation, and the optic nerves were cut

under dim red light. The SCN was quickly dissected and four individual SCNs were pooled to be rapidly frozen on dry ice. Total RNA was extracted by RNeasy mini column (Qiagen, CA, USA). For each time point, 100 ng of total RNA was used as starting material for Affymetrix MOE430 high density arrays (Affymetrix, CA, USA). For *Rorα^Cre^;Lhx1^loxP^* mice, the SCNs were collected every 4 hr. Total RNA was processed for qRT-PCR analyses following standard protocols.

## DNA microarray data analysis

### Identification of circadian transcriptome in the SCN

Data were analyzed by using COSOPT to identify the transcripts that cycle with a time period between 20 and 30 hr, as described previously (*Panda et al., 2002a*). To identify gene families that are overrepresented in the rhythmically expressing genes in the SCN, we used the DAVID pipeline (*Dennis et al., 2003*).

### Identification of the light-regulated genes

Average expression levels of individual probesets were compared between respective light and dark samples. Those showing a fold change of 2 or higher at any of the 9 time-points were further examined.

### Identification of SCN-enriched genes

We used mouse tissue expression database (*Su et al., 2004*) to identify SCN-enriched transcripts. We focused on probesets that had much higher signal in the SCN (in terms of the median temporal expression) than the remaining 82 tissues (duplicate values for each tissue), 14 neural tissues (duplicate), and the hypothalamus. Mathematically, this is interpreted as a high standard deviation for the given probeset expression level in the SCN relative to other tissues. We thus defined two Z values to measure how many standard deviations away the SCN expression is relative to the remainder of the data set, as follows:

Z (all) = ([Median temporal expression in the SCN] − [Average expression across 82 × 2 + 1 tissues])/ (Standard deviation of expression across 82 × 2 + 1 tissues)

Z (neural) = ([Median temporal expression in the SCN] − [Average expression across 2 × 14 + 1 neural tissues])/(Standard deviation of expression across 2 × 14 + 1 neural tissues)

To provide a further layer of stringency, we performed hierarchical clustering of all tissues to identify the anatomical region of the brain that is closest to the SCN in terms of the gene expression signature. This region was the anatomically adjacent hypothalamus. We thus set our third criterion for SCN specificity of a gene as a high value of the SCN/Hypothalamus gene expression values.

We shortlisted probe sets that had Z (all tissues) ≥ 1.63, Z (neural tissues) ≥ 1.44, and SCN/ hypothalamus ≥6. Further stringency was imposed by the requirement that median temporal SCN expression value for the probeset must exceed 200. 230 probe sets meet these criteria. Notably, many of these had previously been described as SCN-specific genes (*Welsh et al., 2010*; *Kasukawa et al., 2011*) and thus validated our approach.

## Quantitative RT-PCR (qRT-PCR)

### One hour light from CT40

*Opn4^Cre/+^;R26^iDTR/+^* and control *Opn4^+/+^;R26^+/+^* mice were individually housed in a wheel running cage and maintained in 12:12 LD cycle. Mice were intraperitoneally injected with diphtheria toxin to ablate melanopsin-expressing RGCs in *Opn4^Cre/+^;R26^iDTR/+^* mice as described (*Hatori et al., 2008*). Ablation of melanopsin RGCs was confirmed by the loss of light entrainment of circadian activity rhythms. In addition to these mice, *rd/rd;Opn4^−/−^*, *Opn4^−/−^* and *rd/rd* mice were maintained in 12:12 LD, then transferred to DD. Mice were exposed to 1 hr light from CT40 or kept in dark, and sacrificed to collect the SCNs at 1 hr from the beginning of light under light or dark.

### Wild type and *Opn4^−/−^* mice under 1 hr light from CT30, 40, or 46

C57BL/6J and *Opn4^−/−^* mice were maintained in LD, then transferred to DD. Mice were exposed to 1 hr light from CT30, 40, or 46 or kept in dark, and sacrificed to collect the SCNs at 0, 1, 2, or 4 hr from the beginning of light under light or dark.

### Measurement of mRNA levels in *Rorα^Cre^;Lhx1^loxP^* mice

*Rorα^Cre/Cre^;Lhx1^loxP/loxP^* and the control mice were kept in LD cycle, then transferred to DD. Mice were collected at CT18, 22, 26, 28, 30, 32, 36, and 40. For circadian gene expression, total RNA from each

time point was subject to qRT-PCR. For non-oscillating transcripts, data from all timepoints were pooled to evaluate levels of expression.

### qRT-PCR

Total RNA from an individual mouse SCN was extracted following RNeasy mini column protocols (Qiagen, CA, USA). cDNA synthesis was carried out with Superscript III (Invitrogen, CA, USA) or qScript cDNA SuperMix (Quanta Biosciences, MD, USA). qRT-PCR was carried out in experimental (at least) triplicates using Power SYBR Green reagent in AB7900HT 384-well system (Applied Biosystems, CA, USA). Primer sequence information is available upon request. Abundance was calculated by normalization to beta Actin (Actb).

### Histology

Procedures for alkaline phosphatase staining for $Rora^{Cre};Z/AP$ brain, X-gal staining for $Rora^{Cre};R26R$ brain, and anterograde tracing with fluorescent cholera toxin subunit B (CTB) have been described previously (*Hatori et al., 2008*; *Chou et al., 2009*; *Brown et al., 2010*).

### Locomotor activity measurement

Daily locomotor activity of mice individually housed in wheel running cages was measured following standard methodology (*Siepka and Takahashi, 2005*). Typically, 6- to 10-week-old mice kept in cages were placed inside light tight boxes with independent illumination. During the light phase, the mice received ~150 lux of white light from fluorescent light source. Wheel running activity in 5 min bins was continuously collected and later analyzed by ClockLab software (Actimetrics, Evanston, IL, USA). All routine animal husbandry care during the dark phase was performed under dim red light.

### Multiunit activity recording of the SCN slices

Mice were sacrificed by cervical dislocation followed by rapid dissection. The brains were cooled down in an ice-cold ACSF solution (125 mM NaCl, 25 mM KCl, 1 mM MgCl$_2$, 1.25 mM NaH$_2$PO$_4$, 2 mM CaCl$_2$, 20 mM Glucose, 26 mM NaHCO$_3$, Penicillin 5000 IU/ml, and Streptomycin 5000 μg/ml) saturated with 95% O$_2$/5% CO$_2$. Coronal slices (~300 μm) were then prepared using a tissue chopper and trimmed to ~10 mm$^2$ slices containing both nuclei. Finally, the slices were transferred to the multi electrodes arrays (MEA). The MEA consists of a glass recording chamber, on the bottom of which are engraved 256 electrodes (10 μm in diameter, situated every 60 μm) and arranged in a 16 × 16 square grid (Multichannel Systems, Reutlingen, Germany). The chamber was continuously perfused with heated (35°C) and oxygenated ACSF-containing antibiotics. Extracellular electrical activity was continuously monitored (signal was acquired from all 256 channels, 10 kHz) and spikes crossing a threshold set at 3 times the standard deviation of the noise on each channel were recorded and stored for off-line analysis. Just prior to placing the MEA on the amplifier, a bright field picture of the slice position on the electrode was rapidly taken to assess SCN placement. The electrodes covering the SCN were then continuously recorded from 2 to 4 days. Channels displaying noise or monotonically decreasing activity were excluded from subsequent analysis. We recorded from animals housed for at least 2 weeks either in DD or LD (12/12) conditions. Animals were handled under dim red light until the optic nerve was cut. Data analysis and display were performed using NeuroExplorer (Plexon Denton, TX), Oriana (Kovach Computing Services, UK), and custom software written in MATLAB (MathWorks, Natick, MA). Peaks of firing were determined after smoothing of data (rloess, MATLAB) and fitting with a sinusoidal function.

Daily administration of VIP was realized by switching the perfusion from the tank containing the medium described earlier to a tank containing the same medium supplemented with VIP (Calbiochem, EMD Millipore, MA) at the concentration of 25 nM for 1 hr.

### Plasmid construction

The coding regions corresponding to full-length mouse and human Lhx1 were amplified by PCR from pineal cDNA and subcloned into pcDNA3.1-TOPO (Invitrogen, CA, USA) to yield expression plasmids mouse Lhx1/pcDNA3.1 and human Lhx1/pcDNA3.1, respectively. Site-directed mutagenesis (Stratagene, CA, USA) was performed to construct mouse Lhx1/pcDNA3.1 to generate a point mutation changing asparagine (amino acid 230) to serine. A DNA fragment corresponding to ~1 kb of mouse *Vip* promoter was amplified by PCR from mouse genomic DNA and cloned into pGL3 basic vector (Promega) to yield the *Vip* reporter vector.

## Transcriptional assay

293T cells were cultured in DMEM supplemented with 10% FBS. The cells (40,000 cells) in 96-well plates were transfected by using TransIT-LT1 (Mirus Bio, WI, USA) with various amounts of expression plasmid (total amount was adjusted to 250 ng by adding empty vector pcDNA3.1), and 5 ng of firefly luciferase reporter plasmid. The cell lysates were prepared 46 hr after the transfection and subjected to dual-luciferase assay by luminometry (Promega, WI, USA).

## Acknowledgements

We thank Sheena Keding and Hiep Le for technical help. This work was partially supported by fellowships from the Japan Society for the Promotion of Science, and Messinger Healthy Living to MH and from Fyssen and Catharina Foundation to LSM. HA and Mary K Chapman Foundation, and The Leona M and Harry B Helmsley Charitable Trust grant #2012-PG-MED002 supported SG. NIH grants NS31558 and MH086147 to DDMO, NS80586 to MG and NIH grant EY016807 and Hearst Foundation to SP funded the study. Funding from Glenn Foundation, P30 CA014195 and P30 EY019005 facilitated circadian activity and gene expression analyses.

Author contributions: MH, SG, LM performed experiments and data analyses, MH, DDMO, MG, and SP conceptualized the experiments, MH and SP wrote the manuscript. Correspondence and requests for materials should be addressed to satchin@salk.edu or mhatori@a6.keio.jp.

## Additional information

### Funding

| Funder | Grant reference number | Author |
| --- | --- | --- |
| National Eye Institute | EY016807 | Satchidananda Panda |
| National Institute of Diabetes and Digestive and Kidney Diseases | DK091618 | Satchidananda Panda |
| National Institute of Neurological Disorders and Stroke | NS31558 | Dennis D M O'Leary |
| National Institute of Mental Health | MH086147 | Dennis D M O'Leary |

The funders had no role in study design, data collection and interpretation, or the decision to submit the work for publication.

### Author contributions

MH, SG, LSM, SP, Conception and design, Acquisition of data, Analysis and interpretation of data, Drafting or revising the article; MG, DDMO'L, Conception and design, Drafting or revising the article, Contributed unpublished essential data or reagents

### Ethics

Animal experimentation: This study was performed in strict accordance with the recommendations in the Guide for the Care and Use of Laboratory Animals of the National Institutes of Health. All of the animals were handled according to approved institutional animal care and use committee (IACUC) protocols (#12-00026) of the Salk Institute for Biological Studies. The protocol was approved by the IACUC committee of the Salk Institute. All surgery was performed under IACUC approved anesthesia, and every effort was made to minimize suffering.

## Additional files

### Supplementary files

• Supplementary file 1. Probesets and corresponding genes showing circadian expression of mRNA abundance in the mouse adult SCN.

• Supplementary file 2. (**A**) Transcripts showing >2 fold expression changes after 1 hr light pulse from CT6, CT16, and CT22. The dark controls for this data set were from *Supplementary file 1*. Therefore, for consistency in naming, the CT6, CT16, and CT22 samples are labeled as CT30, CT40, and CT 46. (**B**) Transcripts showing both light regulation and rhythmic expression. (**C**) Fold changes (ratio of light exposed SCN value to dark control SCN value) for light-modulated transcripts.

• Supplementary file 3. SCN-enriched transcripts and their expression in 83 mouse tissues.

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
