## [Decision Letter]

Thank you for sending your work entitled “Lhx1 maintains synchrony among circadian oscillator neurons of the SCN” for consideration at *eLife*. Your article has been favorably evaluated by a Senior editor and two reviewers, one of whom, Louis Ptáček, is a member of our Board of Reviewing Editors. We are very pleased to inform you that your article has been accepted for publication.

The manuscript by Hatori and colleagues describes an elegant line of experiments showing that Lhx1 is a master regulator of clock neurons in the SCN, and for communication and coordination among these neurons. The experiments range for extensive expression analysis of SCN in WT mice under different lighting regimens. They go on to identify Lhx1 as a candidate of particular interest and then do in vitro and in vivo experiments that support this supposition. There is beautiful correlation of the light induced transcriptional changes with the degree to which there is light-induced behavioral phase shifting.

One concern you might like to address before publication is the use of Lhx1 as ‘THE’ master regulator of SCN. I favor a more conservative approach in referring to it as ‘A’ master regulator.

[Editors’ note: given the minor nature of the revisions, there is not an accompanying Author response.]